# Use of the Phase-Based Model of Smoking Treatment to Guide Intervention Development for Persons Living with HIV Who Self-Identify as African American Tobacco Smokers

**DOI:** 10.3390/ijerph16101703

**Published:** 2019-05-15

**Authors:** Rebecca Schnall, Jasmine Carcamo, Tiffany Porras, Ming-Chun Huang, Monica Webb Hooper

**Affiliations:** 1School of Nursing, Columbia University, New York, NY 10032, USA; jc5207@cumc.columbia.edu (J.C.); tnp2113@cumc.columbia.edu (T.P.); 2School of Engineering, Case Western Reserve University, Cleveland, OH 44106, USA; ming-chun.huang@case.edu; 3Case Comprehensive Cancer Center, Case Western Reserve University, Cleveland, OH 44106, USA; monica.hooper@case.edu

**Keywords:** HIV, tobacco cessation, African American, intervention

## Abstract

Cigarette smoking is highly prevalent among persons living with the human immunodeficiency virus (HIV) (PLWH), with rates as high 50% as compared to 14% in the general U.S. population. Tobacco use causes morbidity and mortality in PLWH, and tobacco-related harm is substantially higher in PLWH than smokers in the general population, providing the scientific premise for developing effective tobacco cessation interventions in this population. To better address this issue, we conducted six focus group sessions with 45 African American smokers who are living with HIV to understand the barriers to smoking cessation and the strategies that would be helpful to overcome these barriers. We organized our findings by the Phase-Based Model of Smoking Treatment to understand the intervention components that are needed at each phase to help PLWH successfully quit smoking. Participants in our focus group sessions articulated key components for incorporation into tobacco cessation intervention for PLWH: a personalized plan for quitting, reminders about that plan, and a support system. Participants thought that their HIV and tobacco use were disassociated. Participants described barriers to the use of pharmacotherapy, including adverse side effects of the gum and patch and concerns about the negative health effects of some oral medications. Substance use was identified as a commonly co-occurring condition as well as a barrier to successfully ceasing to smoke tobacco products. In summary, these findings offer information on the components of a tobacco cessation intervention for PLWH, namely reminders, a support system, substance use treatment, and monitoring to prevent relapse.

## 1. Introduction

Cigarette smoking is highly prevalent among persons living with the human immunodeficiency virus (HIV) (PLWH). The prevalence of smoking in the general U.S. population has gradually declined to 14% in 2017; the lowest rate ever recorded [1]. However, this has not been true for PLWH who have disproportionately high smoking rates (40–70%) [2,3,4]. In our own studies across nearly 450 PLWH in New York City (NYC) [5] and 150 PLWH in Birmingham, AL [6], 50% of our study sample are cigarette smokers.

Tobacco use causes morbidity and mortality in PLWH, and tobacco-related harm is substantially higher in PLWH than smokers in the general population. The high rates of smoking present grave health implications for PLWH, placing them at increased risk for: bacterial pneumonias [7], acute bronchitis and tuberculosis [8,9,10,11,12,13,14,15], early development of emphysema [16,17,18,19,20,21], and lung and cervical cancers at a younger age than the general population [22,23,24,25,26,27]. These disparities in health outcomes are partly due to a higher prevalence of tobacco use in PLWH than the general population and partly attributable to increased susceptibility to the impact of tobacco compared to other smokers. For instance, case-control studies show that acute coronary syndrome is nearly doubled for PLWH who smoked compared to HIV-negative controls [28], and compared to the general population, lung cancer occurs at a younger age and after shorter exposure to cigarettes [29,30].

Evidence-based interventions for tobacco cessation exist, but there are hurdles to their effective use by PLWH. Research shows that, in addition to accelerating the development of adverse health consequences of tobacco use, intervening to improve the rate of smoking cessation has the potential to significantly improve the health and longevity of HIV-positive smokers. Many innovative and effective smoking-cessation treatments, both behavioral and pharmacologic, have been developed over the past several decades [31], but they have largely been aimed at the general population. Therefore, it is not clear whether these treatment strategies are suitable or effective for cohorts with population-specific concerns and clinical issues, such as PLWH [31]. Few tobacco-cessation interventions have been tested among PLWH, and of those which have, there is ‘very low’ quality evidence that tobacco-cessation interventions were effective in the short-term, and ‘moderate’ quality evidence indicating similar outcomes to controls in the long-term [32]. Given the few randomized controlled trials (RCTs) examining smoking-cessation interventions for PLWH, and the major methodological limitations of many of these studies (e.g., lack of randomization, comparison conditions, treatment fidelity assessments, abstinence verification tests), it is critical to develop evidence-based tobacco-cessation interventions to address the complex and unique needs of PLWH (e.g., risk factors, treatment needs).

Guided by the Phase-Based Model (PBM) of Smoking Treatment [33], we sought to better understand the different phases of smoking cessation (i.e., 1. Motivation, 2. Preparation, 3. Cessation, 4. Maintenance) for PLWH, with each of the phases presenting its own challenges and opportunities. We focused on African Americans, as this population is disproportionately affected by HIV [34], and has greater difficulty quitting smoking [35,36]. Building on these findings, we sought to identify the intervention components that would address these phase-specific challenges to improve tobacco cessation.

## 2. Methods

All study activities were approved by the Columbia University Medical Center Institutional Review Board (IRB). Participants were recruited from our study registry of research participants who agreed to be contacted for future use and through study flyers posted at the Columbia-University-affiliated HIV clinic. Inclusion criteria were: PLWH aged 18 and older, able to read and write in English, current smoker, smoked at least 100 cigarettes in their lifetime, seeking cessation treatment, and self-identified as Black/African American. We conducted six focus group sessions with 45 PLWH. Prior to the start of the focus group session, participants completed a demographic survey administered via Qualtrics Software and then watched the Pathways to Freedom: Leading the Way to a Smoke-Free Community© (PTF) video [37]. PTF was designed specifically to address the needs of African American tobacco smokers, and includes culturally specific elements (e.g., emphasis on family, religion/spirituality, history of the tobacco industry and African Americans, menthol cigarette use). Following their viewing of the video, we conducted a focus group session to understand their thoughts and reactions to the video, barriers to smoking cessation, strategies for overcoming these barriers, and specific considerations for PLWH who self-identify as African American tobacco smokers. Focus group sessions lasted between 45 and 60 min and were moderated by the lead author and a co-author (Tiffany Porras). The lead author has extensive experience leading focus group sessions and trained the co-author. The moderator used an interview guide during the focus group sessions. The overall goal of this analysis was to understand the barriers to smoking cessation and the strategies that would be helpful to overcome these barriers for African American smokers who are living with HIV.

## 3. Data Analysis

We audio-recorded and transcribed verbatim all focus group discussions, and analyzed them using a thematic approach. Data were organized by the Phase-Based Model (PBM) of Smoking Treatment [33]. This Model posits that there are different phases of smoking cessation (i.e., 1. Motivation, 2. Preparation, 3. Cessation, 4. Maintenance), each producing its own obstacles and opportunities, and emphasizing the need to identify components that work especially well at each phase [38]. Thus, we sought to understand each of these phases from the perspective of PLWH who self-identify as African American tobacco smokers to guide the development of an intervention designed for this population.

The first author (Rebecca Schnall) reviewed all transcripts and organized them according to the Phases of the Phase-Based Model of Smoking Treatment [33], and the second author (Jasmine Carcamo) independently reviewed all transcripts and the coding of the transcripts. Disagreements on coding were resolved through discussion. The first author then coded all the transcripts by categorizing relevant statements in the transcripts under each theme [39]. The second author then reviewed the coded statements. Disagreements on the codes were resolved through discussion. Another author (Tiffany Porras) then verified the results of the analysis after reading through all transcripts.

## 4. Results

### 4.1. Sample

Forty-five PLWH self-identifying as Black/African-American participated in the smoking cessation focus group sessions and completed the Qualtrics survey. Participants’ ages ranged from 28 to 63 years with a mean age of 52 years (SD 7.78). Twenty-four (53.3%) of the participants were female and 19 (42.2%) were male. Forty-three (95.5%) of the participants self-identified as non-Hispanic and 2 (4.4%) self-identified as Hispanic/Latino. Thirty-one (68.8%) members of the study sample reported their most recent viral load as undetectable (<200 copies/mL). Further details on the sample demographics can be found in Table 1.

### 4.2. Focus Group Findings

Findings from the focus group session are organized by each of the phases of the model. The definition for each phase as well as a sample quote are illustrated in Table 2. Further description of each of these phases in the context of PLWH who smoke are explained below.

### 4.3. Motivation Phase

The *Motivation phase* comprises smokers unwilling to make a quit attempt. Despite our inclusion criterion of people who want to quit smoking, we had one participant who was adamant during the focus group session that she was not interested in quitting smoking. She explained, “I was diagnosed with HIV in 2011. Now for me, it didn’t affect nothing. I go to the doctor. My health is A-1. My doctor is telling me; hey baby, what you doing? Are you still smoking cigarettes? Yeah. I think you need to quit, but I ain’t going to tell you not to. My job, my drug test is cool. I do what I do. But as far as my health, it don’t affect me.” Another woman confided in the group and said, “To be honest with you, I don’t want to quit, even though all of the negative sides there are. I just don’t want to quit. I love that jack pull. I mean I love it and I crave it”.

On the other hand, there were many participants who appreciated the health effects of smoking. One participant said, “So, it’s affecting our immune system, which means we’re more susceptible to things. And under the hood, it’s killing me. I hate stairs and I hate hills.” Another participant agreed and said, “I can’t do steps, either.” Another participant described, “Cigarettes are not good for my skin. It breaks me out every time.” Participants acknowledged that some of their health conditions were directly attributable to their tobacco use. For example, one participant said, “I have COPD [chronic obstructive pulmonary disorder] now. I’m just upscaled from being asthmatic to COPD, and I know I have problems breathing. And, when I don’t have cigarettes, it seems like I breathe better.”

Participants highlighted the exacerbation of negative health effects of smoking for PLWH. One person explained that the video needed to emphasize the urgency of quitting smoking for PLWH. He said, “Well maybe what might be missing…it may be the urgency that people with HIV have, when quitting. Only because the HIV complicates what you’re going to get from smoking anyway. So if you’re likely to get lung cancer, an HIV-infected person is like double.” Participants acknowledged that smoking exacerbated their health conditions, stating, “With HIV and smoking cigarettes, it makes it ten times worse. You can’t have surgeries because you don’t heal properly when you are smoking cigarettes.” Finally, a participant attributed the extent of his pain to smoking and explained, “That pain probably would be more suppressed had I wouldn’t have been smoking cigarettes.”

### 4.4. Preparation Phase

Given our inclusion criteria, most of the study participants were in the *Preparation phase.* In this phase, the smoker is willing to make a quit attempt, but the patient will likely need to use some form of pharmacotherapy to be able to achieve a successful quit attempt. Participants in the focus groups provided suggestions for their peers on pharmacotherapy-based treatment that they heard had worked or they had tried and had been helpful. Participants’ discussion of their use of pharmacotherapy was mostly focused on the *Preparation Phase* since most of our study participants had not quit smoking and were focused on how to use the pharmacotherapy in the future when they decided to stop smoking [40]. For example one woman said, “Ask them for the inhaler, because I’m not trying quit. And I tell you, it works for me. I did it this morning before I came here. And I don’t want one now. It works. I swear.” There were also some concerns about some of these strategies, specifically many barriers to the gum. A number of participants commented that they wear dentures and so they can’t chew gum. Another participant reported trying the gum and had other side effects, specifically stating, “The gum does something too. Chewing the gum, it makes me feel light-headed or something.” One participant in another group explained, “So, they gave me the gum. It tastes like wax. I chewed one piece and then that was it. I didn’t want no more of that.” Finally, one participant explained, “I had the gum before, but I had an allergic reaction.”

In regards to oral medications, such as Chantix, participants also expressed concerns about taking additional medications. One participant said, “And there’s so many side effects to the pill. I take 19 pills a day. I don’t like taking pills every day. Some participants had friends and family who had used Chantix and one reported, “I know Chantix, because my cousin took it. And it worked for her.” But others continued to express concerns about the adverse effects of some of these medications, for example, “The Chantix ain’t going to do nothing, but mess up your liver.” Participants described a number of barriers to the use of patches. One participant explained that, “If you mess around and smoke with the patches on, you’re going to a heart attack.” Another participant did not use the patch because, “I was scared to use the patches because they told if you use the patches, you may get nightmares.” Another participant explained that, “The patch makes me itch.”

### 4.5. Cessation Phase

The *Cessation phase* comprises the immediate post-quit period (~2–4 weeks after the quit day) when the smoker is actively engaged in cessation intervention and striving to become abstinent. The goal is sustained early abstinence and representative challenges include withdrawal symptoms that escalate and typically peak at this time, lapsing, and a brief timeframe for effective intervention. Participants describe their challenges with triggers that are often what results in their failure to cease smoking. For instance, one woman said, “Sometimes when you get around people, you say, oh, give me a puff. You may not ever be thinking about smoking, but if you see somebody, I used to do that. Oh I see you smoking, give me a puff even though it could hurt me I’m already at a point. So, you know, I’m learning to stay away from people who smoke because I will say give me a puff.“ Another participant explained that it is not only seeing people in person, but “when you’re watching those movies, then all of a sudden, they light one up.” Participants reported a number of other triggers, such as use of other substances: “alcohol”, “when I smoke marijuana, then I always want a cigarette to just cool me down”, “Another big trigger for me is the painkillers. I have hurting joints and stuff like that. And the painkillers actually make me want to smoke cigarettes.” and “It’s alcohol and coffee for me.”

Other participants described factors that are specific to their racial and socioeconomic status that make it difficult for them to cease using tobacco. For example, one participant described how tobacco companies take advantage and “they focus on these poor communities, with these damned loosies—bootleg cigarettes.” Another participant who explained that “they’re producing these things because they know if I give this African American person this, and I give this Caucasian person this, I know the African American is going to take it.”

Participants consistently differentiated their tobacco use from their HIV diagnosis. Across focus groups and participants there was a consistent belief that they are not associated with one another despite the much higher rates of smoking in PLWH than the general population. As one participant said, “There’s no relationship for me.” Another participant explained, “I’ve been HIV positive for 30 years. I was a smoker before I got HIV positive. I was drinking. I was smoking weed before I got HIV positive. So I wouldn’t know. It was already there. I was already smoking.” Another participant succinctly discriminated between smoking and living with HIV and stated, “You can put your cigarette down, but you can’t get rid of HIV.”

Participants acknowledged the stress and depression associated with living with HIV and how tobacco use helped alleviate some of these psychological factors. For instance, one participant explained, “I think for me, when I was diagnosed with HIV. I think that enhanced me to want to smoke more…Because of the depression and the stress and all of that.” Another participant stated, “Your point was people who have the virus and tend to have more smokers in that group. I think it’s a stress thing…man, I got this…because I went through a period I forgot I was HIV [positive].”

### 4.6. Maintenance Phase

The *Maintenance phase* follows the establishment of initial abstinence in the *Cessation phase* and is of indeterminate length. The chief goal is the preservation or restoration of abstinence, while representative challenges include flagging motivation, poor adherence to interventions, and the transition of lapses to relapse. A number of participants had previously quit smoking for greater than the immediate post-quit period and described the challenges and encounters that they faced in maintaining tobacco cessation. One man explained, “The most I went was six months. And I walked in the store. And I just said; let me get a pack of Newport. I just walked in the store, six months, just let me get a pack of Newport. The six months that I stopped was the most. And that was in the 80s.” Another participant described that the craving overtook her even after she finally quit and described, “I did so many smoking things and I quit it, then one day I was like, just any little thing, the craving was so strong that I walked to the store and I don’t buy one cigarette, got to buy the whole pack because I’m already messed up.” Stressful life events also make maintenance very challenging as one participant said, “I stopped one time for three years. And you know. So, as soon as one of my siblings passed away, you know, I was at the wake, and had depression, and I picked up a cigarette and smoked.” Finally, one participant who was receiving treatment for a brain tumor did not smoke for 6 months while hospitalized and then when he was discharged “they gave me the cigarettes back. And the lighter. As soon as I came out of the hospital, I lit a cigarette, after not having a cigarette for like six months.”

Participants thought it would be important to have tools to maintain their tobacco cessation. For instance, one participant said that he wanted a sensor to stop him from smoking. He described, “So when you do decide to smoke a cigarette, it says; you need to put this cigarette down.” Related to this suggestion a woman in a different focus group said, “It (the smart watch) needs to have a voice on it so when you do decide to smoke…It can tell you; you need to stop.” Another participant wanted a support system and suggested, “How about a hotline number to call somebody?”

### 4.7. Discussion

Our focus group findings were organized by the Phase-Based Model to better inform the components of an intervention that would be appropriate for PLWH who self-identify as African American tobacco smokers. Given that smokers typically undergo numerous stages through the process of tobacco cessation [41,42], better understanding the barriers and strategies to overcome these challenges at each stage can be useful for designing more effective interventions for PLWH who self-identify as African American tobacco smokers.

Many innovative and effective smoking-cessation treatments, both behavioral and pharmacologic, have been developed for tobacco cessation [31]. However, since smoking cessation efforts have largely been aimed at the general population, it is not clear whether these treatment strategies are suitable or effective for cohorts with population-specific concerns and clinical issues, such as PLWH [31]. Few tobacco-cessation interventions have been tested among PLWH, and of those which have, there is ‘very low’ quality evidence that tobacco-cessation interventions were effective in the short-term, and ‘moderate’ quality evidence indicating similar outcomes to controls in the long-term [32]. Given the few randomized controlled trials (RCTs) examining smoking-cessation interventions for PLWH, and the major methodological limitations of many of these studies (e.g., lack of randomization, comparison conditions, treatment fidelity assessments, abstinence verification tests), it is critical to develop evidence-based tobacco-cessation interventions to address the complex and unique needs of PLWH (e.g., risk factors, treatment needs).

Participants in our focus group sessions articulated key components for incorporation into a tobacco-cessation intervention for PLWH. First, participants stressed the need for a personalized plan for quitting, reminders about that plan, and to stop smoking when someone is about to light-up. Participants thought that their HIV and tobacco use were disassociated and therefore a tobacco-cessation intervention may focus on stress and depression associated with living with HIV, but these may not need to be disease specific to HIV and may relate to any chronic illness. Finally, participants described the need for a support system and how useful it would be to talk to someone who can talk them out of lighting up.

Although there is strong evidence for the effectiveness of pharmacotherapy in supporting tobacco cessation [43,44,45,46], there were a number of barriers to uptake of pharmacotherapy. Participants described how gum interferes with their dentures and leaves a bad taste, along with side effects, such as lightheadedness from gum and other nicotine products, and skin irritation from the patch. Participants were also concerned about using oral medications, since many felt that their liver was already compromised from taking antiretroviral therapy and ultimately did not want to take any additional medications. These findings are not new, since past research, although limited, has shown that many smokers are misinformed about the health risks of nicotine replacement therapy and that these misperceptions impede not only the adoption of nicotine replacement therapy, but also compliance during treatment [47]. In summary, these findings point to the need for education (either standalone or integrated into cessation interventions) regarding misperceptions about smoking cessation aids as well as the unique impact of smoking in the context of HIV.

Importantly, there is strong evidence that tobacco use commonly co-occurs with other substance use. Alcohol use is highly correlated with nicotine dependence in PLWH [48]. Additionally, the rate of smoking when used with other substances (e.g., cannabis, cocaine, Lysergic Acid Diethylamide (LSD), methylphenidate) is significantly increased relative to “sober” smoking rates [49]. Substance users are thought to be more addicted to nicotine than non-abusers as evidenced by studies that have documented that abusers smoke higher-nicotine cigarettes, more cigarettes per day, and/or have increased carbon monoxide, nicotine, and cotinine levels than non-abusers [50]. Further, substance use is a key risk factor for smoking cessation failure [51], which was noted throughout our focus group sessions. Participants largely referred to caffeine and alcohol as being triggers for them to smoke, but there was also reference to other illicit substances (e.g., marijuana and cocaine) and painkillers. This finding suggests that tobacco-cessation interventions for PLWH may require concurrent intervention for alcohol, painkillers, and illegal substances for successful maintenance of tobacco cessation.

## 5. Conclusions

Taken together, these findings offer useful information on the components of a tobacco cessation intervention for African American PLWH who are tobacco smokers, namely reminders, a support system, and monitoring to prevent relapse. Future work should explore the practicality of pharmacotherapy for PLWH who self-identify as African American tobacco smokers given the extensive barriers that were identified in our focus group sessions. Finally, future interventions for tobacco cessation in PLWH will likely need to incorporate substance use treatment as this was identified as a commonly co-occurring condition as well as a barrier to successfully ceasing to smoke tobacco products.

## Figures and Tables

**Table 1 ijerph-16-01703-t001:** Study Sample Demographic Characteristics (*N* = 45).

Characteristic	*n* (%)
Education—highest level of education	
Elementary school	1 (2.2)
High school diploma or equivalent	12 (26.6)
Some high school, no diploma	18 (40)
Some college	11 (24.4)
Associate degree or technical degree	3 (6.6)
Work	
Disabled	16 (35.5)
Retired	2 (4.4)
Unemployed	23 (51.1)
Working full-time	2 (4.4)
Working part-time	4 (8.8)
Working off the books	3 (6.67)
Income	
Less than $10,000	25 (55.5)
$10,000–$19,999	9 (20)
$20,000–$39,000	4 (8.8)
Unknown	2 (1.3)
Sexual Orientation	
Heterosexual	32 (71.1)
Bisexual	4 (8.8)
Homosexual	9 (20)
Relationship Status	
Single	27 (60)
Divorced/Separated	1 (2.2)
In a relationship	15 (33.3)
Legally married or in a registered civil union	2 (4.4)
Smoking	
Regular smoker/e-cigarette user	37 (82.2)
Non-smoker/e-cigarette user	1 (2.2)
Occasional smoker/ e-cigarette user	6 (13.3)
Most recent viral load test	
Undetectable OR < 200 copies/ml	31 (68.8)
Detectable OR ≥ 200 copies/ml	8 (17.7)
Unknown, possibly detectable	3 (6.6)
Unknown, possibly undetectable	3 (6.6)

**Table 2 ijerph-16-01703-t002:** Sample Focus Group Quotes organized by the Phase-Based Model of Smoking Treatment.

Phase	Definition	Sample Quote
Motivation phase	The goal is to increase the rate and probability of successful quit attempts. Representative challenges are low quitting motivation, inadequate coping skills, and high levels of smoking and dependence.	“But as far as my health, it don’t affect me. The only time I started thinking about, because I know how to sing. And the only time I started thinking about, well damn, if I don’t smoke a cigarette, maybe my voice will come back.” “It makes it a lot worse for an HIV person. I just think that it makes it… cigarettes make it a lot worse for a person who already has immune deficiencies.”
Preparation phase	Treatment is used to prepare the smoker for the quit attempt.	“Well look at the good perks about it…Well you’re saving my liver because I’m popping all these pills. I’m going to ruin my liver. Why not go ahead and give me this inhaler? Somebody that told me that’s not trying to quit smoking and she really uses this. It worked. It works, literally. The Chantix ain’t going to do nothing, but mess up your liver. So they were just trying to help you out. Why not just go ahead and use this inhaler? I’ve been using it for a year now. And I swear it works.”
Cessation phase	This comprises the immediate post-quit period (~2–4 weeks after the quit day) when the smoker is actively engaged in cessation intervention and striving to become abstinent. The goal is sustained early abstinence and representative challenges include withdrawal symptoms that escalate and typically peak at this time (triggers).	“But then when I come out here, I started smelling it, and I tried to walk away from it. But now it’s constantly in my face. Then it starts up. I get a pull, to two pulls. Two pulls lead to three, they came out smoking again.” “If you’re a smoker and a drinker, when you have that drink, you’re going to light that cigarette.” “Sometime, have you ever watched a movie, you sit there watching the movie and somebody in the movie light up and you’re like this. You don’t even notice that you’re doing it.”
Maintenance Phase	This follows the establishment of initial abstinence in the Cessation phase and is of indeterminate length. The goal is the preservation of abstinence.	“It all comes down to being accountable to something. Whereas if you’re trying to kind of cold turkey do it yourself, you’re not accountable to anybody but yourself. And we’re really good at lying to ourselves.” “And then, you know, I stopped just not too long ago, and I started back up. I stopped for like, maybe like four weeks. Then I started back a little bit. You know, you get depressions. Certain issues that happen in life, and cigarettes are like a comfort, where it smooths you out”

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
