# Peer review of "Use of the Phase-Based Model of Smoking Treatment to Guide Intervention Development for Persons Living with HIV Who Self-Identify as African American Tobacco Smokers"

_ijerph, 2019, doi:10.3390/ijerph16101703_

Round 1

Reviewer 1 Report

Overall, I am very enthusiastic about this manuscript. The need for tailored smoking cessation interventions among PLWH has been largely unrecognized, making this work quite important. I have a few minor comments:

Abstract

Line 14: "... to understand to understand ..." appears to be a typo

Introduction

There are a number of much more recent papers (and based on nationally representative data) that can be cited regarding the prevalence of smoking among PLWH, including Mdodo et al., 2015; Pacek et al., 2014; and Frazier et al., 2018

Can the authors provide references regarding the prevalence of smoking among PLWH from their own studies?

Methods

Line 65: it appears that the authors left a note to themselves in the paper (cite: https://doi.org...) rather than citing the reference itself using reference software

Discussion

Line 261: Can the authors provide a reference regarding the prevalence of nicotine dependence among PLWH with AUDs?

3rd and 4th paragraphs: To my eye, these findings also point to the need for education (either standalone or integrated into cessation interventions) regarding misperceptions about smoking cessation aids as well as the unique impact of smoking in the context of HIV

Author Response

Review

Response

1.1

Introduction: Line 14: "... to understand to understand   ..." appears to be a typo

We removed the typo.

1.2

There are a number of much more recent papers (and based on   nationally representative data) that can be cited regarding the prevalence of   smoking among PLWH, including Mdodo et al., 2015; Pacek et al., 2014; and   Frazier et al., 2018

We have replaced the old references and now include these more recent   citations.

1.3

Can the authors provide references regarding the prevalence of   smoking among PLWH from their own studies?

We have added 2 citations of work that includes these numbers.

1.4

Methods - Line 65: it appears that the authors left a note to   themselves in the paper (cite: https://doi.org...) rather than citing the   reference itself using reference software

We apologize for this oversight and now added this reference to the   paper

1.5

Discussion: Line 261: Can the authors provide a reference   regarding the prevalence of nicotine dependence among PLWH with AUDs?

We have added this reference: Braithwaite   RS, Fang Y, Tate J, et al. Do Alcohol Misuse, Smoking, and Depression Vary   Concordantly or Sequentially? A Longitudinal Study of HIV-Infected and   Matched Uninfected Veterans in Care. AIDS   and behavior. 2016;20(3):566-572

1.6

3rd and 4th paragraphs: To my eye, these findings also point to   the need for education (either standalone or integrated into cessation   interventions) regarding misperceptions about smoking cessation aids as well   as the unique impact of smoking in the context of HIV

We appreciate this comment and have added this discussion point,   specifically stating “In   summary, these findings point to the need for education (either standalone   or integrated into cessation interventions) regarding misperceptions about   smoking cessation aids as well as the unique impact of smoking in the context   of HIV.”

Reviewer 2 Report

Tobacco smoking is highly prevalent among people with HIV, yet this population has few resources for smoking cessation. Developing smoking cessation interventions specifically for this population may be needed as they are often a difficult to reach population with competing health risks and sociodemographic conditions which may make quitting more difficult. This study evaluated a smoking cessation program designed for African Americans as a strategy for African Americans with HIV through the phase-based model of smoking treatment framework. This study is thoughtful, well-written, and the results are of public health importance. I have only a few minor comments related to editing and some suggestions to improve clarity for the reader.

Minor Comments:

1.     Line 14: “to understand” repeated twice

2.     Line 58: The IRB has not been indicated but simply says “XXX”.

3.     Line 60: Indicating which University and the population this draws from would be helpful for the reader.

4.     Table 1: Should “Retired” be indented? This is perhaps an error as it is not a subcategory of “Disabled”. Same with “Unknown” under Income. There are several other categories that should be reviewed for alignment, as well.

5.     The last quote in Table 2 is missing quotation marks.

6.     It would be helpful for the reader if the author justified why the discussion around medication adherence was in the preparation phase instead of in the cessation phase. Certainly there is some overlap in preparation and cessation (ex: choosing a medication and maintaining use), and some more clarification or consideration by the authors would be helpful.

7.     It is not as clear as it should be from the introduction that the authors are evaluating how this tailored intervention impacts thoughts on cessation through this phase-based model. The first paragraph of the discussion does a better job at making this more clear, but setting up the reader in the introduction would be helpful. It seems this study is a qualitative evaluation of the Pathways to Freedom video through the phase-based framework, but this is not clearly indicated in the introduction. The second paragraph in the discussion (Lines 229-240) actually seems like it would be a helpful component of the introduction.

Author Response

2.1

Line 14: “to understand” repeated twice

We removed the typo.

2.2

Line 58: The IRB has not been indicated but simply says “XXX”.

This has been revised and now states: Columbia   University Medical Center

2.3

Line 60: Indicating which University and the population this   draws from would be helpful for the reader.

We have revised and include “Columbia University”

2.4

Table 1: Should “Retired” be indented? This is perhaps an error   as it is not a subcategory of “Disabled”. Same with “Unknown” under Income.   There are several other categories that should be reviewed for alignment, as   well.

We have revised and revised the other indentations that needed   revision.

2.5

The last quote in Table 2 is missing quotation marks.

We have added the quotation marks

2.6

It would be helpful for the reader if the author justified why   the discussion around medication adherence was in the preparation phase   instead of in the cessation phase. Certainly there is some overlap in   preparation and cessation (ex: choosing a medication and maintaining use),   and some more clarification or consideration by the authors would be helpful.

We have added in the following: Participants’ discussion of their use   of pharmacotherapy was mostly focused on the Preparation Phase since most of our study participants had not   quit smoking and were focused on how to use the pharmacotherapy in the future   when they decided to stop smoking.40

2.7

It is not as clear as it should be from the introduction that   the authors are evaluating how this tailored intervention impacts thoughts on   cessation through this phase-based model. The first paragraph of the   discussion does a better job at making this more clear, but setting up the   reader in the introduction would be helpful. It seems this study is a   qualitative evaluation of the Pathways to Freedom video through the   phase-based framework, but this is not clearly indicated in the introduction.   The second paragraph in the discussion (Lines 229-240) actually seems like it   would be a helpful component of the introduction.

We appreciate this suggestion and have moved it to the introduction   section.